# An Efficient Monte-Carlo Simulation for the Dynamic Reliability Analysis of Jacket Platforms Subjected to Random Wave Loads

Wei Lin [1] and Cheng Su [1,2,*]

1   School of Civil Engineering and Transportation, South China University of Technology, Guangzhou 510640, China; wlin@scut.edu.cn
2   State Key Laboratory of Subtropical Building Science, South China University of Technology, Guangzhou 510640, China
*   Correspondence: cvchsu@scut.edu.cn; Tel.: +86-20-8711-1636

**Abstract:** The growing demand for the application of jacket platforms in deep water requires more attention on the assessment of structural reliability. This paper is devoted to the dynamic reliability analysis of jacket platforms subjected to random wave loads with the Monte-Carlo simulation (MCS), in which a sample size of the order of magnitude of $10^4$ to $10^5$ for repeated time–history analyses is required for small failure probability problems, and a duration time up to three hours needs to be considered in the time–history analyses for a specific sea condition. To tackle the difficulty involved in the MCS, the explicit time-domain method (ETDM) is used for the required time–history analyses of jacket platforms, in which truncated explicit expressions of critical responses with regards to the contributing loading terms are first established and then used for numerous repeated sample analyses. The use of ETDM greatly enhances the computational efficiency of MCS, making it feasible for the dynamic reliability analysis of jacket platforms under random wave loads. A jacket platform with 11,688 degrees of freedom was analyzed for the evaluation of dynamic reliability under a given sea condition, indicating the accuracy and efficiency of the present approach and its feasibility to practical structures.

**Keywords:** jacket platform; system dynamic reliability; first-passage failure criterion; explicit time-domain method; Monte-Carlo simulation



## 1. Introduction

Jacket platforms have been widely used in the exploitation of offshore oil and gas. A reliability analysis is of great concern to the design of jacket platforms due to the uncertainties inevitably involved in wave loads and structural resistances [1,2]. Traditional static methods for the reliability analysis based on the design-level strength are frequently used, such as the first-order and second-order reliability methods [3–7]. However, as wave loads are dynamic in nature, the static analysis methods are inadequate to account for the dynamic effects associated with jacket platforms, and the first-passage dynamic reliability analysis is of interest in practical applications.

Over the past few decades, the power spectrum method (PSM) based on the equivalent linearized Morison equation for wave drags [8,9] was frequently adopted for the first-passage dynamic reliability analysis of jacket platforms, in which certain assumptions regarding the probability distribution of the level-crossing number, e.g., the Poisson process assumption or the Markov process assumption, need to be considered [10,11]. Furthermore, such level-crossing process-based methods can only obtain an approximate solution for component reliability, and for the system reliability problems involving different component failure modes, only the upper and lower bounds of the failure probability can be estimated [12,13].

The Monte-Carlo simulation (MCS) is the most versatile method to implement the first-passage dynamic reliability analysis of jacket platforms under random wave loads, in which the more accurate nonlinear Morison equation can be directly taken into account for wave drags. An accurate solution to the component reliability can be obtained without additional assumptions regarding the probability distribution of level crossings. Furthermore, for the system reliability problems, the partial correlation properties among different failure modes can be automatically reflected in the process of MCS, leading to accurate results of the system failure probability without using the upper and lower bounds formula. However, as the failure probabilities of jacket platforms are generally set to be adequately small, large sample sizes are required for accurate estimations of the reliability with the MCS, leading to enormous computational effort. In particular, for the dynamic reliability analysis of jacket platforms under a stationary sea condition, a typical duration time of wave loads up to three hours needs to be considered, which makes it even more difficult to apply the MCS in ocean engineering practice [14,15].

The importance sampling (IS) and subset simulation (SS) techniques are often employed to reduce the sample size of the MCS for the assessment of small failure probability. However, the IS technique requires information about the failure region for the construction of the sampling distribution function, which is difficult to achieve in high-dimensional reliability problems [16–18]. On the other hand, the SS technique converts the original problem into the estimation of a sequence of large conditional probabilities [19,20], in which conditional samples from a specially designed Markov chain need to be generated and the generation method has a significant influence on the solution accuracy [21]. As an alternative strategy to enhance the efficiency of the MCS, the Constrained NewWave model reduces the duration time of wave loads by using statistical methods to predict the most probable highest-wave surface elevation within a short time duration of hundreds of seconds [22–27]. Nevertheless, certain assumptions regarding the probability distribution of the amplitude of the wave surface elevation are required, which inevitably result in additional errors for the dynamic reliability analysis.

In this paper, the system dynamic reliability analysis of the jacket platforms subjected to nonlinear random wave loads is conducted with the MCS based on the explicit time-domain method (ETDM) [28–30]. Owing to the explicit formulation of the dynamic responses in terms of random loads at different time instants, there is no need to repeatedly solve the equation of motion of the structure for different sample analyses. Furthermore, in view of the long duration time of the wave loads involved in the dynamic reliability analysis of the jacket platforms, truncated explicit expressions of critical responses with contributing loading terms are further adopted for the truncated sample analyses. The use of ETDM greatly enhances the computational efficiency of the MCS for the dynamic reliability analysis of jacket platforms under random wave loads. An engineering example involving a jacket platform with 11,688 degrees of freedom (DOFs) is analyzed for the dynamic reliability under a given sea condition, indicating the accuracy and efficiency of the present approach and its feasibility in practical structures.

## 2. Equation of Motion

The equation of motion for a jacket platform subjected to wave loads can be expressed as

$$M\ddot{U}(t) + C\dot{U}(t) + KU(t) = LF(t) \tag{1}$$

where $M$ and $C$ are the total mass and damping matrix of the system, respectively; $K$ is the stiffness matrix of the structure; $U(t)$, $\dot{U}(t)$ and $\ddot{U}(t)$ are the nodal displacement, velocity and acceleration vector of the structure, respectively; and $L$ is the orientation matrix of the concentrated wave loading vector $F(t)$.

In Equation (1), the total mass matrix $M$ consists of two parts and can be expressed as

$$M = M_0 + M_A \tag{2}$$

where $M_0$ and $M_A$ are the structural mass matrix and the added mass matrix induced by the accelerated sea water, respectively. The added mass matrix $M_A$ can be obtained through the distributed added mass $m_A = \frac{1}{4}C_A\rho\pi D^2$ of a cylinder in sea water, in which $C_A$ is the added mass coefficient [31–33], and $\rho$ and D are the water density and the diameter of the cylinder, respectively.

The total damping matrix $C$ in Equation (1) also consists of two parts, which can be expressed as

$$C = C_0 + C_H \tag{3}$$

where $C_0$ and $C_H$ are the structural damping matrix and the hydrodynamic damping matrix induced by the blockage of sea water, respectively. The hydrodynamic damping matrix $C_H$ can be determined by towing tank tests using the small-scale model of the jacket platform [34].

The concentrated wave loading vector $F(t)$ in Equation (1) can be expressed as

$$F(t) = F_I(t) + F_D(t) \tag{4}$$

where $F_I(t)$ and $F_D(t)$ are the concentrated inertial force and drag force vector, respectively, which can be determined through the distributed inertial force vector $f_I(t)$ and the distributed drag force vector $f_D(t)$ acting on the cylinder in sea water. According to the nonlinear Morison equation [35], they can be expressed as

$$f_I(t) = \begin{Bmatrix} f_{Ix} \\ f_{Iy} \\ f_{Iz} \end{Bmatrix} = K_M \begin{Bmatrix} \dot{v}_{Nx} \\ \dot{v}_{Ny} \\ \dot{v}_{Nz} \end{Bmatrix}, \quad f_D(t) = \begin{Bmatrix} f_{Dx} \\ f_{Dy} \\ f_{Dz} \end{Bmatrix} = K_D|v_N| \begin{Bmatrix} v_{Nx} \\ v_{Ny} \\ v_{Nz} \end{Bmatrix} \tag{5}$$

in which $K_M = \frac{1}{4}C_M\rho\pi D^2$ and $K_D = \frac{1}{2}C_D\rho D$, with $C_M$ and $C_D$ being the inertia and drag coefficient, respectively, and $v_N = \begin{bmatrix} v_{Nx} & v_{Ny} & v_{Nz} \end{bmatrix}^T$ and $\dot{v}_N = \begin{bmatrix} \dot{v}_{Nx} & \dot{v}_{Ny} & \dot{v}_{Nz} \end{bmatrix}^T$ are the fluid–particle velocity and acceleration vector normal to the cylinder, respectively, with the superscript T denoting the transposition of matrix, and $|v_N| = \sqrt{v_{Nx}^2 + v_{Ny}^2 + v_{Nz}^2}$.

The distributed inertial force vector $f_I(t)$ and the distributed drag force vector $f_D(t)$ in Equation (5) are illustrated in Figure 1, in which the x-direction refers to the direction of the wave propagation, which is supposed to be the same as the direction of the in-line steady current, and the z-direction refers to the vertical direction with the origin O located at the still water level. $P(x, y, z)$ represents an arbitrary point on a cylinder member $C_1C_2$ with inclined angles with respect to the x-, y- and z-directions denoted as $\alpha_x$, $\alpha_y$ and $\alpha_z$, respectively. For clarity, the current–particle velocity $v_c$ along the x-direction, the wave–particle velocities $v_x$ and $v_z$ and the wave–particle accelerations $\dot{v}_x$ and $\dot{v}_z$ along the x- and z-directions for a two-dimensional swell model are also shown in Figure 1. Then, the fluid–particle velocity vector normal to the cylinder, $v_N$, in Equation (5) can be obtained as

$$v_N = \begin{Bmatrix} v_{Nx} \\ v_{Ny} \\ v_{Nz} \end{Bmatrix} = \begin{bmatrix} 1 - \cos^2\alpha_x & -\cos\alpha_x\cos\alpha_z \\ -\cos\alpha_x\cos\alpha_y & -\cos\alpha_y\cos\alpha_z \\ -\cos\alpha_x\cos\alpha_z & 1 - \cos^2\alpha_z \end{bmatrix} \begin{Bmatrix} v_c + v_x \\ v_z \end{Bmatrix} \tag{6}$$

It can be seen from Equations (4)–(6) that the nonlinear wave loads can be determined using the current–particle velocity $v_c$, the wave–particle velocities $v_x$ and $v_z$ and the wave–particle accelerations $\dot{v}_x$ and $\dot{v}_z$. It is worth noting that, in this study, the influence of the current–particle velocity has been taken into consideration in the wave loads.

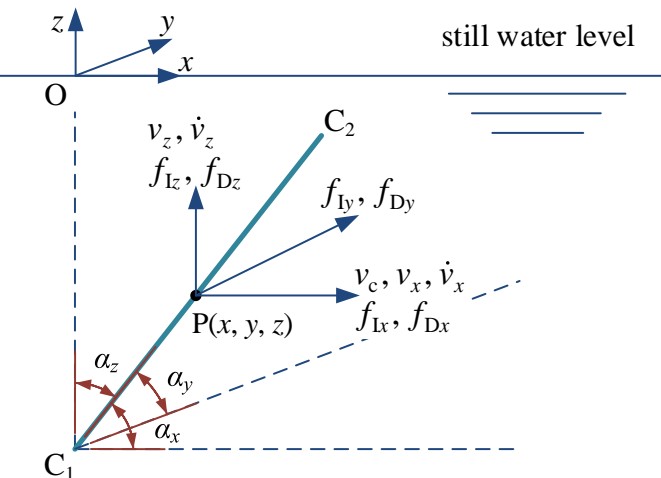

**Figure 1.** Distributed wave loads and fluid–particle velocities and accelerations.

### 3. Explicit Formulation of Structural Responses

*3.1. Explicit Expressions of Structural Responses*

Define the state vector of the jacket platform as $V(t) = \left[ U(t)^{\mathrm{T}} \ \dot{U}(t)^{\mathrm{T}} \right]^{\mathrm{T}}$. Solve the equation of motion shown in Equation (1) with a specific numerical integration scheme, in which, without loss of generality, assume $V_0 = V(0) = 0$ and $F_0 = F(0) = 0$. Then, the explicit expression of the state vector can be derived as [30]

$$V_i = A_{i,1}F_1 + A_{i,2}F_2 + \cdots + A_{i,i-1}F_{i-1} + A_{i,i}F_i \quad (i = 1, 2, \cdots, n) \tag{7}$$

where $n$ denotes the number of time steps for the time–history analysis; $V_i = V(t_i)$ and $F_j = F(t_j)$ $(1 \le j \le i \le n)$, in which $t_i = i\Delta t$ and $t_j = j\Delta t$, with $\Delta t$ being the time step; and $A_{i,j}(1 \le j \le i \le n)$ are the coefficient matrices expressed in the closed forms as

$$\begin{cases} A_{1,1} = Q_2, \quad A_{2,1} = TQ_2 + Q_1, \quad A_{i,1} = TA_{i-1,1} \ (3 \le i \le n) \\ A_{i,j} = A_{i-1,j-1} \ (2 \le j \le i \le n) \end{cases} \tag{8}$$

in which $T$, $Q_1$ and $Q_2$ can be derived through the Newmark-$\beta$ integration scheme and expressed as

$$\begin{cases} T = \begin{bmatrix} H_{11} & H_{12} \\ H_{21} & H_{22} \end{bmatrix}, \quad Q_1 = \begin{bmatrix} R_1 \\ R_3 \end{bmatrix} L, \quad Q_2 = \begin{bmatrix} R_2 \\ R_4 \end{bmatrix} L \\ H_{11} = \hat{K}^{-1}\left(S_1 - S_3 M^{-1} K\right), \quad H_{12} = \hat{K}^{-1}\left(S_2 - S_3 M^{-1} C\right) \\ H_{21} = a_3(H_{11} - I) + a_5 M^{-1} K, \quad H_{22} = a_3 H_{12} - a_4 I + a_5 M^{-1} C \\ R_1 = \hat{K}^{-1} S_3 M^{-1}, \quad R_2 = \hat{K}^{-1}, \quad R_3 = a_3 R_1 - a_5 M^{-1}, \quad R_4 = a_3 R_2 \\ \hat{K} = K + a_0 M + a_3 C \\ S_1 = a_0 M + a_3 C, \quad S_2 = a_1 M + a_4 C, \quad S_3 = a_2 M + a_5 C \\ a_0 = \frac{1}{\beta \Delta t^2}, \quad a_1 = \frac{1}{\beta \Delta t}, \quad a_2 = \frac{1}{2\beta} - 1, \quad a_3 = \frac{\gamma}{\beta \Delta t}, \quad a_4 = \frac{\gamma}{\beta} - 1, \quad a_5 = \frac{\Delta t}{2}\left(\frac{\gamma}{\beta} - 2\right) \end{cases} \tag{9}$$

where $\mathbf{I}$ denotes the unit matrix, and $\beta$ and $\gamma$ are the two parameters associated with integration accuracy and stability. In this paper, $\beta = 0.25$ and $\gamma = 0.50$ are adopted for the unconditionally stable integration scheme.

From the point of view of engineering applications, only certain critical responses are required for the system dynamic reliability analysis of jacket platforms. Suppose that $s(t)$ is the response vector comprised of the critical responses. With the explicit formulation

of $V_i$ shown in Equation (7), the explicit expression of the critical response vector can be obtained as

$$s_i = \phi V_i = a_{i,1}F_1 + a_{i,2}F_2 + \cdots + a_{i,i-1}F_{i-1} + a_{i,i}F_i \quad (1 \leq i \leq n) \tag{10}$$

where $s_i = s(t_i)$, $\phi$ is the response transfer matrix and the coefficient matrices $a_{i,j}$ for $s_i$ can be obtained from $A_{i,j}$ as

$$a_{i,j} = \phi A_{i,j} \quad (1 \leq j \leq i \leq n) \tag{11}$$

Note that, when the specific critical response in $s$ represents the displacement or velocity of a certain DOF, the corresponding row in $\phi$ consists of 0 and 1, and when the specific critical response in $s$ represents an element force component, the corresponding row in $\phi$ contains the entries in element force–displacement matrix.

The substitution of Equation (8) into Equation (11) yields

$$\begin{cases} a_{i,1} = \phi A_{i,1} \quad (1 \leq i \leq n) \\ a_{i,j} = a_{i-1,j-1} \ (2 \leq j \leq i \leq n) \end{cases} \tag{12}$$

It can be observed from the above equation that a recursive relationship exists among the coefficient matrices $a_{i,j}(1 \leq j \leq i \leq n)$.

### 3.2. Truncated Explicit Expressions of Structural Responses

It can be seen from Equation (10) that the coefficient matrix $a_{i,m}$ represents the influence of $F_m$ at time $t_m$ on the critical response vector $s_i$ at time $t_i$. From the physical point of view, for a damped structure, $F_m$ will exert little influence on $s_i$ when $t_m$ is sufficiently far away from $t_i$, leading to the norm $\|a_{i,m}\|$ approaching zero. Therefore, to further enhance the computational efficiency, Equation (10) can be truncated as

$$s_i = a_{i,m+1}F_{m+1} + a_{i,m+2}F_{m+2} + \cdots + a_{i,i-1}F_{i-1} + a_{i,i}F_i \quad (1 \leq i \leq n) \tag{13}$$

when

$$\frac{\|a_{i,m}\|}{\|a_{i,i}\|} \leq \varepsilon \tag{14}$$

where $\varepsilon$ denotes the truncation index and can be taken as a small value, e.g., $\varepsilon = 10^{-3}$. Obviously, only a number of ($i$—$m$) contributing loading terms are kept in Equation (13).

Note that, if Equation (14) cannot be satisfied until $m = 0$, with $\|a_{i,0}\|$ defined as 0, it means that Equation (10) cannot be truncated for the current time instant, and in this case, Equation (13) actually turns back into Equation (10).

Using the recursive relationship shown in Equation (12), one can rewrite Equations (13) and (14) in the following forms:

$$s_i = a_{i-m,1}F_{m+1} + a_{i-m-1,1}F_{m+2} + \cdots + a_{2,1}F_{i-1} + a_{1,1}F_i \quad (1 \leq i \leq n) \tag{15}$$

when

$$\frac{\|a_{i-m+1,1}\|}{\|a_{1,1}\|} \leq \varepsilon \tag{16}$$

It should be noted that, as compared with the original explicit formulation of the critical responses shown in Equation (10), the explicit formulation shown in Equation (15) represents the truncated calculations of the critical responses, which is particularly suitable for the response time–history analysis of the complex jacket platforms in consideration of a long duration time of random wave loads up to three hours for a specific sea condition. Furthermore, it can be observed from Equation (15) that, among the coefficient matrices shown in Equation (12), only $a_{1,1}$, $a_{2,1}$, $\cdots$, $a_{i-m-1,1}$ and $a_{i-m,1}$ need to be stored for the truncated formulation of the critical structural responses.

## 4. System Reliability Analysis with MCS Based on ETDM

### 4.1. Simulation of Wave–Particle Velocities and Accelerations

In the MCS, the wave–particle velocities and accelerations, which are generally modeled as zero-mean stationary Gaussian random processes, need to be generated for determination of the drag and inertial forces via Equations (5) and (6). Within the framework of linear random wave theory [36], the wave–particle velocities and accelerations shown in Figure 1 can be simulated using the spectral representation method [37,38], which can be expressed as follows:

$$v_x(x,z,t) = \sum_{i=1}^{n_\omega} \sqrt{2S_{v_x}(z,\omega_i)\Delta\omega_i} \cdot \cos(\kappa_i x - \omega_i t + \varepsilon_i) \tag{17}$$

$$v_z(x,z,t) = \sum_{i=1}^{n_\omega} \sqrt{2S_{v_z}(z,\omega_i)\Delta\omega_i} \cdot \sin(\kappa_i x - \omega_i t + \varepsilon_i) \tag{18}$$

$$\dot{v}_x(x,z,t) = \sum_{i=1}^{n_\omega} \sqrt{2S_{\dot{v}_x}(z,\omega_i)\Delta\omega_i} \cdot \sin(\kappa_i x - \omega_i t + \varepsilon_i) \tag{19}$$

$$\dot{v}_z(x,z,t) = -\sum_{i=1}^{n_\omega} \sqrt{2S_{\dot{v}_z}(z,\omega_i)\Delta\omega_i} \cdot \cos(\kappa_i x - \omega_i t + \varepsilon_i) \tag{20}$$

where $S_{v_x}(z,\omega)$ and $S_{v_z}(z,\omega)$ are the power spectrum density functions of the wave–particle velocity along the $x$- and $z$-directions, respectively; $S_{\dot{v}_x}(z,\omega)$ and $S_{\dot{v}_z}(z,\omega)$ are the power spectrum density functions of the wave–particle acceleration along the $x$- and $z$-directions, respectively; $n_\omega$ is the number of representative frequencies; $\omega_i$ and $\Delta\omega_i$ ($i = 1, 2, \cdots, n_\omega$) are the $i$th representative frequency and the corresponding frequency interval, respectively; $\kappa_i$ is the wave number of the $i$th cosine wave; and $\varepsilon_i$ is the random phase angle of the $i$th cosine wave with uniform distribution in $[0, 2\pi]$.

The above power spectrum density functions of wave–particle velocities and accelerations can be expressed as

$$S_{v_x}(z,\omega) = H_z^2(z,\omega)S_\eta(\omega) \tag{21}$$

$$S_{v_z}(z,\omega) = H'^2_z(z,\omega)S_\eta(\omega) \tag{22}$$

$$S_{\dot{v}_x}(z,\omega) = \omega^2 H_z^2(z,\omega)S_\eta(\omega) \tag{23}$$

$$S_{\dot{v}_z}(z,\omega) = \omega^2 H'^2_z(z,\omega)S_\eta(\omega) \tag{24}$$

where $S_\eta(\omega)$ is the power spectrum density function of the wave surface elevation, and $H_z(z,\omega)$ and $H'_z(z,\omega)$ are the depth-dependent functions, expressed as

$$H_z(z,\omega) = \frac{\omega\cosh[\kappa(z+h)]}{\sinh(\kappa h)} \tag{25}$$

$$H'_z(z,\omega) = \frac{\omega\sinh[\kappa(z+h)]}{\sinh(\kappa h)} \tag{26}$$

where $h$ is the water depth of the sea site, and $\kappa$ and $\omega$ should satisfy the following dispersion relationship:

$$\omega^2 = g\kappa \cdot \tanh(\kappa h) \tag{27}$$

where $g$ is the gravitational acceleration.

### 4.2. System Limit-State Function

For the system dynamic reliability evaluation of the jacket platforms subjected to random wave loads, the failure event is generally defined as the event that the von Mises stress of any structural member exceeds the yielding stress of the steel material [1,39].

Therefore, the system reliability of the jacket platforms can be modeled as a series system reliability problem. Assuming that the system failure mode consists of $n_{\mathrm{f}}$ component failure modes, the system limit-state function can be defined as

$$Z(t_{\mathrm{d}}) = \min_{q=1,2,\cdots,n_{\mathrm{f}}} Z_q(t_{\mathrm{d}}) \tag{28}$$

where $t_{\mathrm{d}}$ is the duration time of random loads, and $Z_q(t_{\mathrm{d}})$ $(1 \leq q \leq n_{\mathrm{f}})$ denotes the $q$th component limit-state function.

Based on the first-passage failure criterion, the component limit-state function can be defined as

$$Z_q(t_{\mathrm{d}}) = \frac{f_{\mathrm{y},q}}{\max\limits_{0<t\leq t_{\mathrm{d}}} \overline{\sigma}_q(t)} - 1 \quad (1 \leq q \leq n_{\mathrm{f}}) \tag{29}$$

where $\overline{\sigma}_q(t)$ $(1 \leq q \leq n_{\mathrm{f}})$ is the von Mises stress that controls the $q$th component failure mode, and $f_{\mathrm{y},q}$ is the yielding stress of the steel material for the $q$th component.

The substitution of Equation (29) into Equation (28) yields the system limit-state function as

$$Z(t_{\mathrm{d}}) = \min_{1\leq q\leq n_{\mathrm{f}}} \left[ \frac{f_{\mathrm{y}}}{\max\limits_{0<t\leq t_{\mathrm{d}}} \overline{\sigma}_q(t)} - 1 \right] \tag{30}$$

The above system limit-state function corresponds to the reliability problem of series systems. For parallel systems or series–parallel-coupled systems, the system limit-state function can be found in reference [40].

### 4.3. ETDM-Based MCS for System Reliability Analysis

To enhance the efficiency of the MCS, the truncated ETDM with Equation (15) is incorporated into MCS for repetitive time–history analyses of the jacket platforms subjected to wave loads. This hybrid approach is termed as ETDM-based MCS, and the solution procedure for the system dynamic reliability analysis of the jacket platforms using this approach is summarized as follows:

(1) Determine the system failure event for the jacket platforms subjected to random wave loads and the corresponding system limit-state function $Z(t_{\mathrm{d}})$ by Equation (30), in which the critical response vector $\boldsymbol{s}$ can be determined.

(2) Calculate the matrices $\boldsymbol{T}$, $\boldsymbol{Q}_1$ and $\boldsymbol{Q}_2$ by Equation (9), and according to the selected critical response vector s, determine the coefficient matrices $\boldsymbol{a}_{1,1}$, $\boldsymbol{a}_{2,1}$, $\cdots$, $\boldsymbol{a}_{i-m-1,1}$, $\boldsymbol{a}_{i-m,1}$ in Equation (15) by Equations (8) and (12).

(3) Generate a sufficient number of samples of wave–particle velocities and accelerations by Equations (17)–(20) and determine the corresponding samples of the nonlinear wave loading vector $\boldsymbol{F}^k(t)$ $(k = 1, 2, \cdots, M)$ by Equations (4)–(6), in which $M$ denotes the number of samples.

(4) For the $k$th sample of the wave loading vector $\boldsymbol{F}^k(t)(1 \leq k \leq M)$, calculate the critical stress vector $\boldsymbol{s}_i^k$ $(1 \leq i \leq n)$ by Equation (15), and then, determine the corresponding von Mises stresses for the critical sections of the structural members.

(5) Determine the value of the system limit-state function $Z^k(t_{\mathrm{d}})$ by Equation (30), and check whether $Z^k(t_{\mathrm{d}}) < 0$. If yes, a first-passage failure event occurs. Assume that a total number of $M_{\mathrm{f}}$ failure events occur among the $M$ sample cases. Then, when $M$ is sufficiently large, the system failure probability can be obtained as $P_{\mathrm{f}} = M_{\mathrm{f}}/M$.

It can be seen from Step (2) that the coefficient matrices required in Equation (15) for the explicit formulation of the structural responses need to be calculated just once and can be used for repetitive time–history analyses corresponding to all the sample cases involved in the MCS. This feature leads to a significant reduction in the computational cost compared with the traditional MCS, in which the equation of motion shown in Equation (1) needs to be solved for each sample case. In addition to this, by using Equation (15), the truncated calculations of the critical responses can be easily conducted, which further enhances the

computational efficiency of the MCS for the dynamic reliability analysis of the complex jacket platforms in consideration of the three-hour storm condition.

## 5. Engineering Example

### 5.1. Jacket Platform and Sea Condition

In this section, the dynamic reliability evaluation of a steel jacket platform subjected to random wave loads for a specific sea condition is conducted with ETDM-based MCS. The height of the jacket platform is 86.80 m, and the depth of water is 62.00 m. The jacket platform is modeled with three-dimensional beam elements, and the finite element (FE) model is shown in Figure 2, which consists of 2303 beam elements and 1964 nodes, leading to a total number of 11,688 DOFs. The structural and hydrodynamic damping ratios are taken as 2% and 8%, respectively [34].

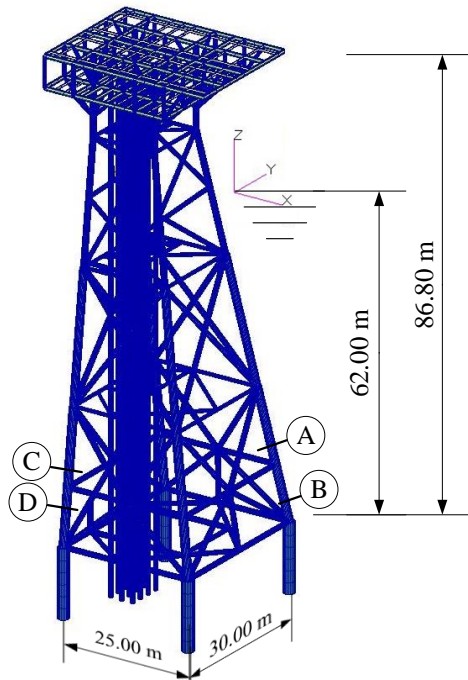

**Figure 2.** The finite element (FE) model of the jacket platform.

The power spectrum density function of the wave surface elevation is assumed to be the two-parameter Pierson–Moskowitz wave spectrum, expressed as [15]

$$S_\eta(\omega) = \frac{5}{16} \frac{\omega_P^4}{\omega^5} H_{Se}^2 \exp\left[-\frac{5}{4}\left(\frac{\omega_P}{\omega}\right)^4\right] \tag{31}$$

$$H_{Se} = [1.0 + 0.5 \exp(-h/25)] H_S \tag{32}$$

where the significant wave height $H_S$ and the spectral peak frequency $\omega_P$ of the wave surface elevation are taken as 11.50 m and 0.44 rad/s, respectively, for a 100-year return period; $H_{Se}$ denotes the effective significant wave height based on the modified Wheeler stretching method in consideration of the influence of the wave loads above the still water level [41]; and $h$ is the water depth of the sea site.

The corresponding power spectrum density functions of the wave–particle velocities and accelerations can be obtained through Equations (21)–(24). The wave–particle velocities and accelerations are then generated with the spectral representation method based on Equations (17)–(20), in which 935 nonuniformly distributed representative frequencies [42] are considered in the range of 0.26 to 2.80 rad/s, with the minimum frequency interval being $5.15 \times 10^{-4}$ rad/s for the duration time of $t_d = 10,800$ s (3 h), according to reference [15].

The generated samples of wave–particle velocities and accelerations at the still water level ($z = 0$) are depicted in Figures 3 and 4, respectively. The simulated spectra of the wave–particle velocities and accelerations at $z = 0$ can be obtained based on 1000 samples of wave–particle velocities and accelerations, which are presented in Figures 5 and 6. For comparison, the target spectra of the wave–particle velocities and accelerations at $z = 0$, determined by Equations (21)–(24) and Equations (31) and (32), are also depicted in Figures 5 and 6. It can be observed from Figures 5 and 6 that the simulated spectra are in good agreement with the target spectra.

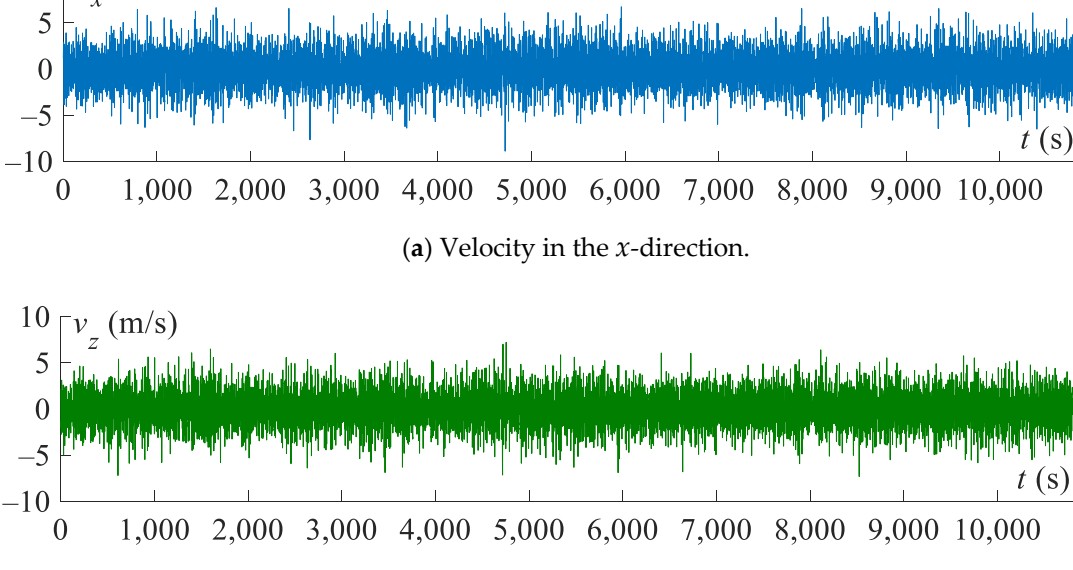

(**a**) Velocity in the $x$-direction.

(**b**) Velocity in the $z$-direction.

**Figure 3.** Samples of wave–particle velocities at a still water level ($z = 0$).

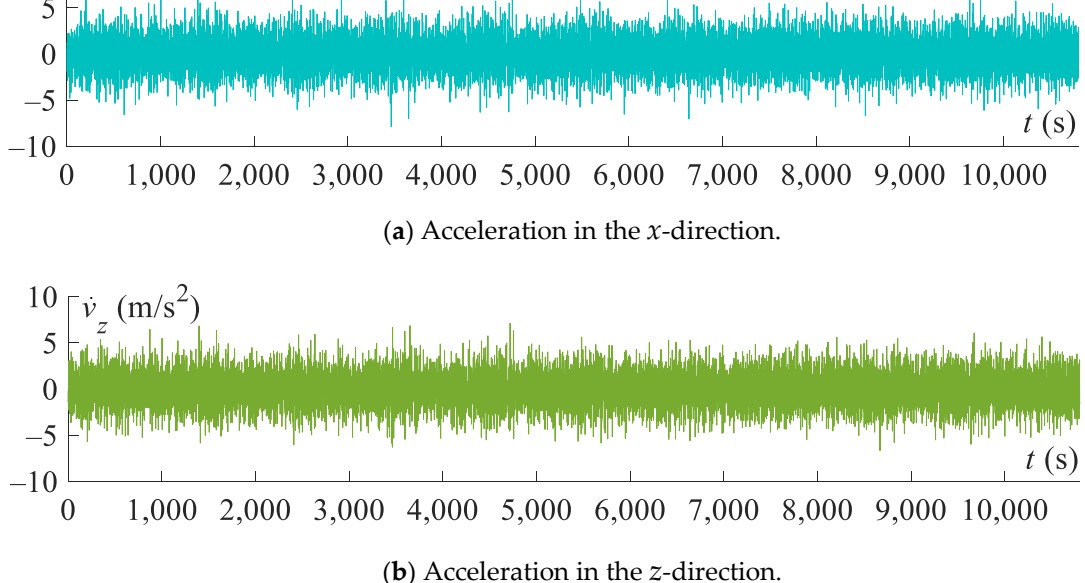

(**a**) Acceleration in the $x$-direction.

(**b**) Acceleration in the $z$-direction.

**Figure 4.** Samples of wave–particle accelerations at a still water level ($z = 0$).

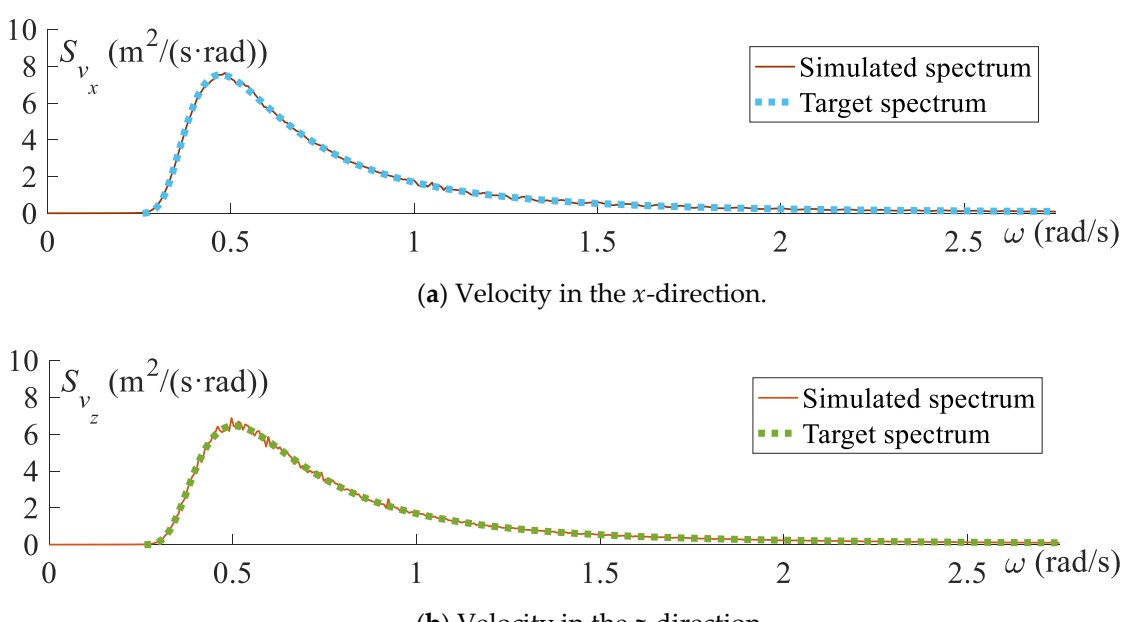

(**a**) Velocity in the *x*-direction.

(**b**) Velocity in the *z*-direction.

**Figure 5.** Power spectrum density functions of wave–particle velocities at a still water level ($z = 0$).

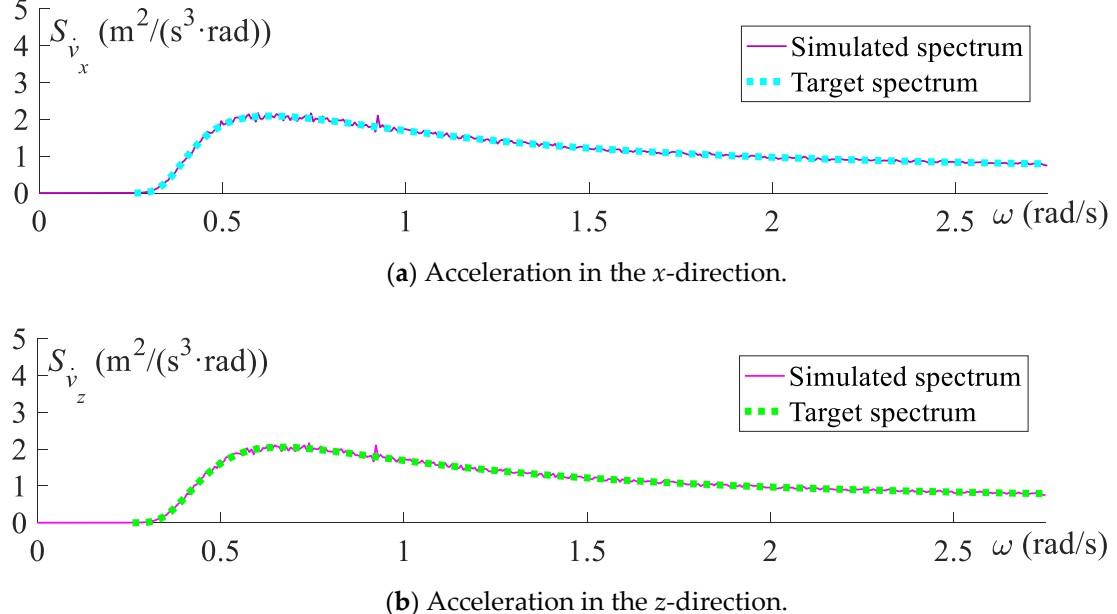

(**a**) Acceleration in the *x*-direction.

(**b**) Acceleration in the *z*-direction.

**Figure 6.** Power spectrum density functions of wave–particle accelerations at a still water level ($z = 0$).

In addition to the wave–particle velocities, the current–particle velocities are also required in Equation (6), which can be determined based on the current profile, given as [15]

$$v_c(z) = v_{c,tide}(0)\left(\frac{h+z}{h}\right)^{1/7} + \frac{1}{2}v_{c,wind}(0)\left[1 + \text{sgn}\left(\frac{50+z}{50}\right)\right] \tag{33}$$

where $v_{c,tide}(0) = 0.514$ m/s and $v_{c,wind}(0) = 0.50$ m/s are the tide current velocity and the wind-generated current velocity at the still water level ($z = 0$), respectively; and sgn($\cdot$) denotes the sign function.

To obtain the wave loads from the fluid–particle velocities and accelerations, the hydrodynamic coefficients involved in Equation (5) are set as $C_M = 2.00$ and $C_D = 1.30$ [34].

Besides, to obtain the added mass matrix in Equation (2), the added mass coefficient is taken as $C_A = C_M - 1 = 1.00$ [34].

### 5.2. Three-Hour-History Analysis of Jacket Platform with ETDM

Under the action of the three-hour wave–particle velocities and accelerations, among which, the samples for $z = 0$ are shown in Figures 3 and 4, the nonlinear wave loads for the jacket platform can be obtained by Equations (4)–(6), in which the influence of the current–particle velocities determined by Equation (33) is also included. The time–history analysis of the critical responses of the jacket platform is carried out using the truncated ETDM with Equation (15). For comparison, two truncation indices, $\varepsilon = 5 \times 10^{-3}$ and $5 \times 10^{-2}$, are adopted in the truncated ETDM. To validate the accuracy of the present approach, the direct Newmark-$\beta$ method is also used for the time–history analysis of the jacket platform subjected to the same wave loads. In the above analysis, the duration time of the wave loads is taken to be $t_d = 10,800$ s (3 h) for the specified sea condition, and the time step is taken as $\Delta t = 0.20$ s, leading to a total of 54,000 time steps.

For critical sections B and D of the battered legs of the jacket platform, shown in Figure 2, the time histories of the normal stresses and shear stresses are depicted in Figures 7 and 8, respectively, from which it can be seen that the results obtained by the truncated ETDM with $\varepsilon = 5 \times 10^{-3}$ are in good agreement with those obtained by the Newmark-$\beta$ method, indicating the accuracy of the present approach. However, certain discrepancies can be observed for the results obtained by the truncated ETDM with $\varepsilon = 5 \times 10^{-2}$, which means that the truncation index $\varepsilon = 5 \times 10^{-2}$ chosen is larger than required for the truncated ETDM.

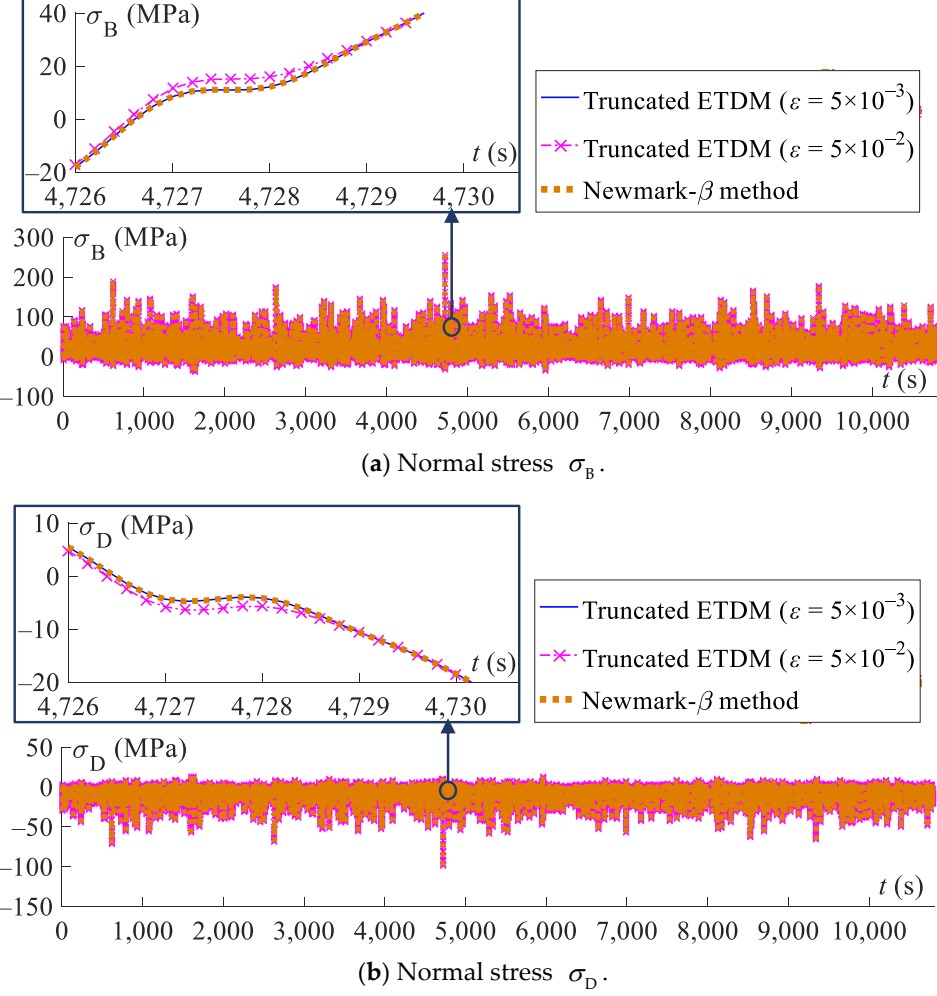

(**a**) Normal stress $\sigma_B$.

(**b**) Normal stress $\sigma_D$.

**Figure 7.** Normal stresses at sections B and D of the battered legs of the jacket platform.

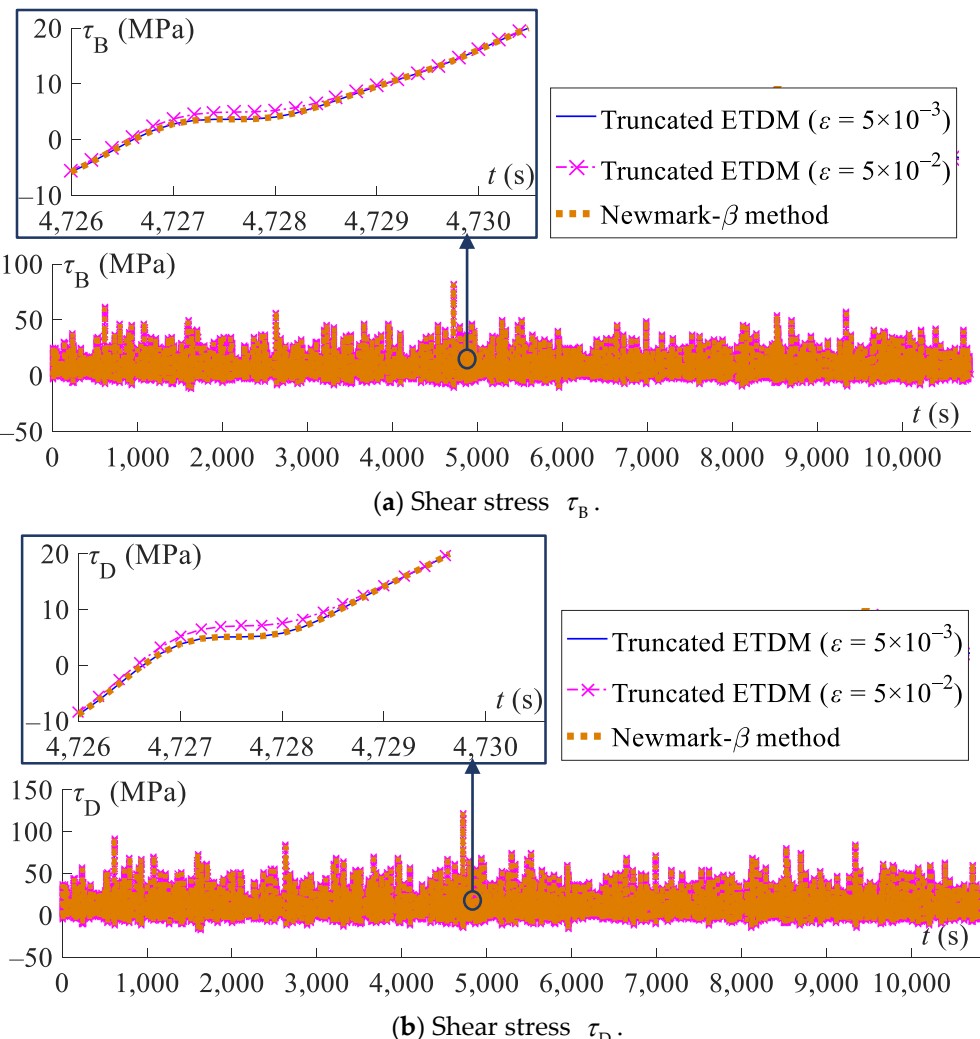

**Figure 8.** Shear stresses at sections B and D of the battered legs of the jacket platform.

The computation times for the truncated ETDM and the Newmark-$\beta$ method are shown in Table 1, from which it can be seen that the truncated ETDM has a much higher efficiency than the direct Newmark-$\beta$ method. It should be noted that, for the truncated ETDM with $\varepsilon = 5 \times 10^{-3}$, according to Equation (16), at most, only 92 contributing loading terms need to be considered in Equation (15) for the calculations of the responses at different time instants from $t_1 = 0.2$ s to $t_{54,000} = 10,800$ s, while, for the nontruncated ETDM, the numbers of the terms in Equation (10) range from 1 to 54,000, corresponding to the responses at time $t_1$ to $t_{54,000}$. This indicates that a large number of terms were omitted in Equation (15) while the high accuracy of the method remained, which is the major reason for the high computational efficiency observed in the truncated ETDM.

**Table 1.** Computation time for different methods. ETDM: explicit time-domain method.

| Method | Elapsed Time (s) |
|---|---|
| Truncated ETDM | 56.45 |
| Newmark-$\beta$ method | 2980 |

Note: All the above computations were done on a PC with an Intel(R) Xeon(R) Platinum 8160 CPU with a 2.10-GHz processor and 256-GB RAM.

### 5.3. Dynamic Reliability Analysis of Jacket Platform with ETDM-Based MCS

The ETDM-based MCS presented in Section 4.3 was used to conduct the system dynamic reliability analysis of the jacket platform under nonlinear random wave loads. It was found that large von Mises stresses occurred at sections A, B, C and D of the battered legs, which are shown in Figure 2. Therefore, for this example, it was supposed that failure occurred once the von Mises stress of any one of the above four sections exceeded the yielding stress of the steel material, which was taken as $f_y$ = 235 MPa for sections A and C and $f_y$ = 350 MPa for sections B and D.

Under the three-hour sea condition described in Section 5.1, different sample sizes of the MCS were first investigated to obtain the convergent system failure probability of the structure. The system failure probabilities obtained with different sample sizes are shown in Figure 9. It can be observed from Figure 9 that the convergent result $P_f$ = 0.37% can be achieved with a sample size of $M$ = 30,000, which satisfies the well-recognized condition $M \geq 100/P_f$ [43]. In what follows, similar investigations will be conducted to determine the sample sizes for different levels of failure probabilities.

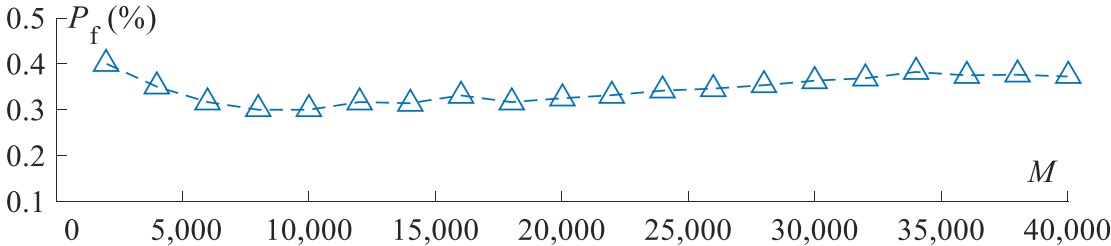

**Figure 9.** System failure probabilities with different sample sizes ($t_d$ = 3 h).

To investigate the influence of the duration time on the reliability analysis, the system failure probabilities $P_f$ with different duration times are evaluated and shown in Figure 10, from which it can be seen that the system failure probability increases with the increase of the duration time. Therefore, for a typical three-hour sea condition, it is necessary to consider a duration time of $t_d$ = 3 h for the dynamic reliability analysis of the jacket platforms.

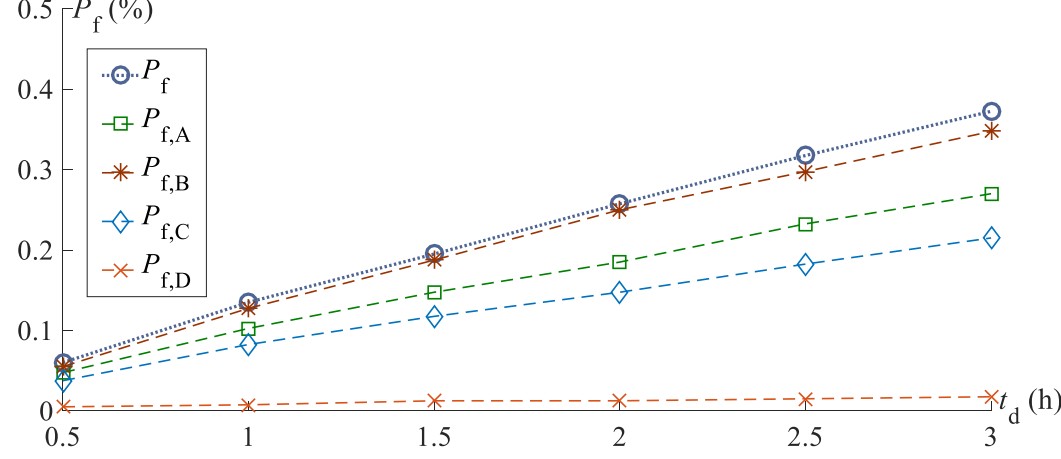

**Figure 10.** Component and system failure probabilities with different duration times.

In addition to the system failure probability $P_f$, the component failure probabilities $P_{f,A}$ to $P_{f,D}$, corresponding to critical sections A to D, are also depicted in Figure 10, from which it can be seen that the component failure probabilities are different from each other and are smaller than the system failure probability. This indicates that the component

reliability analysis will underestimate the failure probability level of the jacket platform, and therefore, the system reliability analysis is necessary for the reasonable evaluation of the failure probability.

In this example, for the system dynamic reliability analysis of the jacket platform subjected to the specified three-hour sea condition, the computation time of the ETDM-based MCS is presented in Table 2. The present method first takes 56 s to calculate the coefficient matrices for the truncated explicit expressions of the critical responses shown in Equation (15). Then, based on Equation (15), it takes 13,500 s to carry out a total of 30,000 sample analyses in the MCS, leading to an extremely short computation time of 0.45 s for each sample analysis on average. The total computation time of the present approach is 13,556 s (approximately 3.8 h), which can be accepted beyond question for such a practical engineering problem. Nevertheless, if we use the direct Newmark-$\beta$ method for the time–history analysis, it will take 2980 s for a single sample analysis, as shown in Table 1, and obviously, the method can hardly be used in the MCS for a sample size of $M = 30,000$.

**Table 2.** Computation time of the ETDM-based Monte-Carlo simulation (MCS).

| Calculation of Coefficient Matrices in Equation (15) (s) | MCS Based on Equation (15) ($M = 30,000$) (s) | Total (s) |
|---|---|---|
| 56 | 13,500 | 13,556 |

Note: All the above computations were done on a PC with an Intel(R) Xeon(R) Platinum 8160 CPU with a 2.10-GHz processor and 256-GB RAM.

## 6. Conclusions

An efficient ETDM-based MCS was developed for the evaluation of the first-passage system dynamic reliability of jacket platforms subjected to a specific three-hour sea condition. To avoid repeatedly solving the equation of motion for numerous time–history analyses of the jacket platform under different samples of wave loads, the explicit formulation of the structural state vector is first established with regards to the wave loads at different time instants. To further enhance the computational efficiency for the time–history analysis with a three-hour duration time of wave loads, truncated explicit expressions of critical responses with a very limited number of contributing loading terms were constructed, which could then be used throughout the whole process of MCS. A jacket platform with 11,688 DOFs was used as an example for the dynamic reliability analysis under a given three-hour sea state, which indicated the accuracy and efficiency of the present approach and its feasibility to practical engineering problems.

**Author Contributions:** Conceptualization, C.S.; investigation, W.L.; methodology, C.S.; validation, W.L.; writing—original draft, Wei Lin and writing—review and editing, C.S. All authors have read and agreed to the published version of the manuscript.

**Funding:** The research is funded by the National Natural Science Foundation of China (51678252) and the Science and Technology Program of Guangzhou, China (201804020069).

**Institutional Review Board Statement:** Not applicable.

**Informed Consent Statement:** Not applicable.

**Data Availability Statement:** Data available in a publicly accessible repository.

**Conflicts of Interest:** The authors declare that they have no conflicts of interest.

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
