# Peer review of "An Efficient Monte-Carlo Simulation for the Dynamic Reliability Analysis of Jacket Platforms Subjected to Random Wave Loads"

_jmse, doi:10.3390/jmse9040380_

Round 1

Reviewer 1 Report

The work seems to be very interesting and it has a potential to be published. I would have few remarks:

  1. In the text, there is no distinguishing between the accelerations and velocities (no evidence of dots). Must be corrected.
  2. It starts with line 110, naming incident forces, it is wrong and it should be inertial forces
  3. My main concern is about the calculation of added mass. You have it constant, but you do calculation over a wide wave spectra. Added mass is wave frequency dependant and must be taken into consideration. But this will completely brake up you hypothesis of having initial matrices constant. How can you defend your approach? How is dynamic reliability analysis affected? aCn you provide some estimates of difference.

Reviewer 2 Report

The manuscript suggests an efficient use of the Monte-Carlo simulation to analyse jacket platforms and get an initial overview of the structures' reliability.

Overall it's a well-written paper. I will recommend editing the abstract to avoid subjective wording. On line 12: remove "huge" sample size. Please suggest a number based on objective information. Ln. 13. These simulations are not "super-long". They are standard 3-6 hour simulations due to certain reasons. Please use objective wording and expressions.

Figure 6. There seems to be a persisting problem with the calculation of values around 1 rad/s. Could you please explain?

Ln. 297, the authors assume 8% as hydrodynamic damping. Please explain why this value was calculated as a percentage while it can be estimated as a summation using the Morison equation.

Round 2

Reviewer 1 Report

Look fine. Can proceed with publishing.